# Nucleation Points: The Forgotten Parameter in the Synthesis of Hydrogel-Coated Gold Nanoparticles

**DOI:** 10.3390/polym13030373

**Published:** 2021-01-26

**Authors:** Adolfo Sepúlveda, Audrey Picard-Lafond, André Marette, Denis Boudreau

**Affiliations:** 1Département de Biochimie, de Microbiologie et de bio-Informatique, Université Laval, Québec, QC G1V 0A6, Canada; adolfo-javier.sepulveda-san-martin.1@ulaval.ca; 2Centre D’optique, Photonique et Laser (COPL), Université Laval, Québec, QC G1V 0A6, Canada; audrey.picard-lafond.1@ulaval.ca; 3Département de Chimie, Université Laval, Québec, QC G1V 0A6, Canada; 4Département de Médecine, Faculté de Médecine, Institut Universitaire de Cardiologie et de Pneumologie de Québec (IUCPQ) et Institut sur la Nutrition et les Aliments Fonctionnels (INAF), Université Laval, Québec, QC G1V 0A6, Canada; andre.marette@criucpq.ulaval.ca

**Keywords:** nanoparticles, core-shell colloids, gold nanoparticles, poly(N-isopropylacrylamide) (pNIPAM), hydrogel, seeded precipitation polymerization

## Abstract

The implementation of gold-hydrogel core-shell nanomaterials in novel light-driven technologies requires the development of well-controlled and scalable synthesis protocols with precisely tunable properties. Herein, new insights are presented concerning the importance of using the concentration of gold cores as a control parameter in the seeded precipitation polymerization process to modulate—regardless of core size—relevant fabrication parameters such as encapsulation yield, particle size and shrinkage capacity. Controlling the number of nucleation points results in the facile tuning of the encapsulation process, with yields reaching 99% of gold cores even when using different core sizes at a given particle concentration. This demonstration is extended to the encapsulation of bimodal gold core mixtures with equally precise control on the encapsulation yield, suggesting that this principle could be extended to encapsulating cores composed of other materials. These findings could have a significant impact on the development of stimuli-responsive smart materials.

## 1. Introduction

In recent years, the design and fabrication of core-shell plasmonic nanomaterials possessing hybrid properties, e.g., the plasmonic properties of noble metal nanoparticles [1] and the volume phase transition behavior of external-stimulated microgels [2], have gained increasing attention towards developing novel light-driven technologies such as sensing [3], photovoltaics [4], and plasmon-mediated photothermal therapies [5], among others [6,7,8]. The encapsulation of metallic nanoparticles within hydrogels is an important component for the fabrication of these hybrid nanomaterials [9], with most reported protocols employing a seeded precipitation polymerization process where functionalized spherical gold nanoparticles [10,11,12,13,14,15,16] are coated with cross-linked poly(N-isopropylacrylamide) (pNIPAM) hydrogel, the latter having a well documented lower critical solution temperature (LCST) of 32–33 °C in water [17]. Understanding the factors governing the encapsulation process is a prerequisite in developing scalable synthesis protocols for these plasmonic-polymer nanocomposites, and several studies have shown that the modulation of certain parameters, such as monomer feed concentration [13], cross-linker density [15], and sequential polymerization [18], may control the hydrogel shell thickness. However, only a few works have addressed the parameters driving the encapsulation of gold nanoparticles. As an example, Rauh et al. have shown that the hydrophobicity of gold seeds is the only factor governing the successful formation of core-shell nanoparticles, and that the encapsulation yield, in principle, is influenced by the overall gold particle surface [19]. However, since this study was performed for a single core diameter, not much is known about the effect of metallic core size or even polydispersity on the polymerization process. In fact, it is not clear whether the encapsulation yield of gold nanoparticles in this type of hybrid architectures is modulated by the overall particle surface or by the number of particles (nucleation points) used in the seeded precipitation polymerization process. Understanding this is critical to develop scalable polymerization protocols for hybrid nanomaterials with tunable properties, including optical properties and shell thickness. 

This work aims to bring new insights concerning the role of nucleation points in the fabrication of gold-hydrogel core-shell nanoparticles and their usefulness as a control parameter over features such as encapsulation yield, particle size and shrinkage capacity of the hybrid nanomaterial. To this end, the encapsulation of spherical gold nanoparticles 15, 35, and 50 nm in diameter was performed by seeded precipitation polymerization of N-isopropylacrylamide (NIPAM) and N,N′-methylenebisacrylamide (BIS). Moreover, bimodal samples were also prepared using mixtures of distinct core sizes in a single one-pot polymerization synthesis.

## 2. Materials and Methods

### 2.1. Chemicals

Gold(III) chloride trihydrate (HAuCl_4_; Sigma-Aldrich, Oakville, ON, Canada, ≥99.9%), sodium citrate tribasic dihydrate (SCTD; Sigma-Aldrich, Oakville, ON, Canada, ≥99.9%), sodium dodecyl sulfate (SDS; Sigma-Aldrich, Oakville, ON, Canada, 95%), butenylamine hydrochloride (B-en-A; Sigma-Aldrich, Oakville, ON, Canada, 97%), N,N′-methylenebisacrylamide (BIS; Sigma-Aldrich, Oakville, ON, Canada, 99%), and potassium persulfate (KPS; Sigma-Aldrich, Oakville, ON, Canada, 99%) were used as received. N-isopropylacrylamide (NIPAM; Sigma-Aldrich, Oakville, ON, Canada, 97%) was purified by recrystallization in n-hexane prior to use. Nanopure water (18 MΩ) was used in all experiments. Glassware and magnetic stirrers used for the syntheses were cleaned with aqua regia and thoroughly rinsed with water.

### 2.2. Synthesis and Functionalization of Gold Nanoparticles

Spherical gold nanoparticles (AuNPs) were prepared by a seed-growth procedure using gold seeds as a starting material to grow larger nanoparticles. This synthesis was adapted from existing protocols [20,21,22] and provides nanoparticles with well-controlled and uniform sizes from 15 to 50 nm in diameter. Briefly, in a clean 250 mL round bottom flask equipped with a magnetic stirring bar and a reflux condenser, 100 mL of 0.23-mM aqueous HAuCl_4_ was brought to a boil using an oil bath at 120 °C under vigorous stirring (900 rpm). In total, 5 mL of 32 mM aqueous SCTD was quickly added to the reaction vessel and left to stir for 30 min before cooling down to room temperature. The resulting 15 nm gold seed particles were diluted to 100 mL with nanopure water. The 35 and 50 nm gold nanoparticles were obtained through a seed-growth procedure performed in successive steps. To begin, 75 mL of nanopure water and 2 mL of 34 mM aqueous SCTD were added to a 250 mL round bottom flask and placed under reflux with vigorous stirring. This was followed by the addition of 10 mL of the 15 nm stock solution and 1 min later of 1.7 mL of 5.9 mM aqueous HAuCl_4_ and left to stir for 45 min. This was followed by a growth step where 1.7 mL of 5.9 mM aqueous HAuCl_4_ and 2 mL of 34 mM aqueous SCTD were added into the vessel and stirred for another 45 min. At this point, the size of the particles was approximately 35 nm in diameter. To obtain 50 nm particles, the growth step was repeated twice before letting the solution cool down to room temperature.

The gold nanoparticles were functionalized with B-en-A to increase their hydrophobicity and promote the precipitation polymerization on their surface [19]. This was done by adding different volumes (depending on core diameter) of 1 mM aqueous SDS to 100 mL of nanoparticle stock solution followed, after 20 min of continuous stirring at room temperature, by the dropwise addition of specific volumes of 1.4 mM B-en-A in ethanol (see Appendix A for details). After an additional 20 min of stirring, the functionalized gold nanoparticles were centrifugated for 14 h at 1000 rcf (relative centrifugal force) and redispersed in 2 mL of nanopure water. The concentration of the particles (expression as particles/mL) was determined by nanoparticle tracking analysis (NTA).

### 2.3. Seeded Precipitation Polymerization

The encapsulation of gold cores was performed using a seeded precipitation polymerization protocol adapted from literature [19]. Briefly, in a 50 mL three-neck round-bottom flask equipped with a magnetic stirring bar and a reflux condenser, 57 mg of NIPAM and 12 mg of BIS (15 mol% relative to NIPAM) were dissolved in 23 mL of nanopure water at room temperature under continuous stirring (600 rpm). Upon complete dissolution, the mixture was heated to 70 °C and purged with nitrogen to remove oxygen. After 20 min, varying volumes (depending on the desired encapsulation yield) of functionalized gold nanoparticles were added dropwise to the solution. It is worth noting that the final volume of the solution mixture containing the functionalized nanoparticles, NIPAM and BIS was kept constant at 25 mL to have the same monomer and cross-linker concentration in each experiment. After 15 min of equilibration time, the polymerization was initiated by the rapid addition of 1 mL of 1.85 mM aqueous KPS and allowed to proceed for 2 h. The solution was cooled to room temperature and the resulting gold-hydrogel core-shell nanoparticles were purified by three consecutive centrifugations at 6000 rcf until obtaining a clear supernatant. Finally, the particles were dispersed in 5 mL of nanopure water and stored at 4 °C.

### 2.4. Characterization

The geometry and size distribution of gold-pNIPAM core-shell nanoparticles were determined by transmission electron microscopy (TEM; Model Tecnai G2 Spirit, FEI, Hillsboro, OR, USA) at an accelerating voltage of 100 kV. Samples were prepared by drop-casting a dispersion of particles on a carbon-coated copper TEM grid (200 mesh, Electron Microscopy Sciences, Hatfield, PA, USA) and the substrate was air-dried prior to measurements. Image analysis was performed using ImageJ software. The size distribution and number concentration of starting gold cores were measured by nanoparticle tracking analysis (NTA; Model NanoSight NS300, Malvern Instruments, Worcestershire, United Kingdom), employing a 532 nm laser module and an external syringe pump to allow a continuous flow rate of 15 μL/min of particle suspension during the measurements. The NTA instrument was calibrated with commercial calibration beads (0.2, 0.5, and 0.76 μm in diameter) from Bangs Laboratories (Fishers, IN, USA). Dynamic light scattering (DLS; Model Zetasizer NanoZS, Malvern Instrument, Worcestershire, United Kingdom) was used to determine the hydrodynamic diameter and polydispersity index of the core-shell nanoparticles. 

## 3. Results and Discussion

Eleven distinct polymerizations were conducted by modulating both the number and size of nucleation points while keeping the volume constant (Table 1). The total number of nucleation points employed for each sample was measured by NTA, while the overall particle surface was determined considering the particle size distribution obtained from TEM images and the number of gold particles in each sample. The successful encapsulation yield of pNIPAM-encapsulated gold cores was calculated from the processing of TEM images. The first five polymerizations (series S1) were carried out with 50 nm gold cores increasing in number from 2.9 to 14.5 × 10^12^ nanoparticles in the reaction volume. As depicted in Figure 1, the resulting encapsulation yield of 50 nm AuNPs@pNIPAM linearly increases with the augmentation of nucleation points added to the polymerization solution. This tunability in the amount of encapsulated gold cores varies from 16% to 94%. Using similar amounts of nucleation points as those of the S1 series, further polymerizations were performed by employing 30 and 15 nm gold nanoparticles as cores, named as S2: 35 nm AuNPs@pNIPAM-X and S3: 15 nm AuNPs@pNIPAM-X, respectively. In terms of encapsulation yields, equivalent results were obtained as those for 50 nm gold nanoparticles at similar particle concentration (Figure 1). The discrepancy in the percentage determined for sample S3: 15 nm AuNPs@pNIPAM-1 compared to samples S1 and S2 with the same number of nucleation points (2.9 × 10^12^) was attributed to a problem in the centrifugation step. Importantly, as shown on the TEM images of every sample (see Appendix A), only spherical microgels containing a single gold core are observed, with no signs of isolated (non-polymerized) gold nanoparticles.

As a further proof of the dominance of the number of nucleation points over overall particle surface on the encapsulation process, two additional polymerization experiments were conducted with bidisperse gold cores. To this end, one of the polymerizations contained a combination of 50 and 65 nm gold nanoparticles (S4: 50/65 nm AuNPs@pNIPAM), while the other one had 25 and 65 nm gold cores (S5: 25/65 nm AuNPs@pNIPAM). The total number of gold cores employed in bimodal samples S4 and S5 is presented in Table 1, whereas the proportion of each gold nanoparticle size can be found in the samples section of the Appendix A. The achieved encapsulation yields of pNIPAM-coated bidisperse gold nanoparticles perfectly match the linear tendency observed with the other samples for the given number of nucleation points, i.e., 37% for sample S4 (6.0 × 10^12^ NPs) and 94% for sample S5 (14.6 × 10^12^ NPs), as shown in Figure 1. Furthermore, the representative TEM images of samples S4 and S5 (depicted in Figure 2) show the successful encapsulation of gold nanoparticles with no signs of aggregation. The greater asymmetry of core-shell particles with the larger cores observed for sample S5 can be attributed to the impact of an excessive centrifugation time to obtain a clear supernatant considering the two distinct sizes of gold nanoparticles in the same solution. To the best of our knowledge, this is the first report showing hydrogel encapsulation onto two-sizes gold cores through the same polymerization process and with tight control over sample morphology.

These results clearly indicate that the encapsulation yield of gold-pNIPAM core-shell nanoparticles synthesized via seeded precipitation polymerization varies according to the number of nucleation points and not by the available overall surface of functionalized-gold cores. In fact, when comparing the yields obtained with the different gold core sizes but controlling their number, it is noticed that, for a given encapsulation yield, the overall particle surface does not have any effect on this parameter, as shown in Table 1 and Appendix A. The latter highlights (with green-dashed rectangles) the three experiments where the number of nucleation points was kept constant but employing different gold core sizes (S1-1, S2-1; S1-2, S2-2; and S1-5, S3-2, S5), resulting in similar encapsulation yields despite the difference in the overall particle surface for each sample. It is worth mentioning that the lack of linearity on the overall particle surface of the S1 series is due to the difference in the particle size distribution between the employed gold nanoparticles batches for each polymerization (Appendix A). This finding, linking the encapsulation ratio to the number of nucleation points rather than the available particle surface, is consistent with the mechanism controlling the precipitation polymerization, where pure microgel particles form due to a homogeneous nucleation procedure. For given synthesis conditions, only a certain number of precursor polymer particles may take shape after the precipitation of growing oligomer and polymer chains [23]. Therefore, the addition of hydrophobic gold nanoparticles [19] into the reaction mixture allows the formation of these precursor polymer particles onto the gold core surface. In this sense, if the number of surface-functionalized gold nanoparticles is close to the number of intrinsically formed precursor polymer particles, solely gold-pNIPAM core-shell nanoparticles are formed. Otherwise, both hydrogel-encapsulated gold nanoparticles and microgels without metallic cores form with a yield depending on the number of cores initially incorporated.

To study the effect of the number of nucleation points on the size and shrinkage capacity of the core-shell nanoparticles, DLS experiments were performed at 15 and 50 °C to measure their hydrodynamic diameters in the swollen and shrunken states, respectively. To compare the influence of metallic cores within the hydrogel network, pure microgel spheres were synthesized under the same polymerization conditions and used as a reference (see Appendix A). Given the difference in the hydrogel shell thickness coating with the bidisperse gold cores of samples S4 and S5, these samples were not considered in this study to avoid any bias in the measurements. Table 2 summarizes the mean hydrodynamic diameters of the hybrid nanoparticles measured at both temperatures and their shrinkage capacity (shrinking ratio). As observed, there is a direct dependency between the hydrodynamic diameters and the number of nucleation points utilized in the polymerizations. More precisely, increasing the number of nucleation points leads to a decrease in the diameters of core-shell nanoparticles measured at both temperatures, e.g., the augmentation of nucleation points of the sample group S1 resulted in the modulation of the core-shell particle size from 501 to 231 nm (at 15 °C). This trend agrees with what has been already observed in the encapsulation of gold cores of 14 nm [19].

From the DLS measurements, the analyzed particles presented a low polydispersity index (PDI < 0.1) even on experiments with lower encapsulation percentages (see Appendix A). These results suggest that, for a given experiment, both the gold-hydrogel core-shell nanoparticles and the coreless hydrogel particles have similar sizes. Indeed, the similar PDI values for all samples of the S1 series indicate do not show a clear tendency to state that those experiments with fewer hydrogel-encapsulated gold cores (and therefore more pure hydrogel particles) are more polydisperse compared to those having a higher encapsulation ratio. Accordingly, the shrinking ratio (see Appendix A) was calculated in consideration of the mean hydrodynamic radius of each sample to ascertain their shrinkage capacity as a function of the number of nucleation points. This value provides information on the ability of the material to decrease in size from the swollen to the collapsed state. As displayed in Figure 3, the synthesized pure pNIPAM microgels show a higher shrinking ratio value than those of pNIPAM-coated gold nanoparticles, whereas the different core-shell samples have similar shrinking ratio values at a given number of nucleation points regardless of the gold core size. The high shrinking ratio value of pure pNIPAM microgels means that they can shrink up to 40% of their original size. This low shrinkage capacity compared to the gold-hydrogel core-shell nanoparticles may be attributed to the inhomogeneous cross-linker distribution effect along the microgel produced by the quicker consumption of the BIS molecules relative to pNIPAM throughout the polymerization process [24]. The difference in the consumption kinetics of these two compounds leads to a core-shell type microgel with a densely cross-linked domain in the interior and a lower dense cross-linked domain toward the surface [25,26]. Dulle et al. have shown by sequential semi-batch polymerizations that the hydrogel network homogeneity plays a role in the shrinkage capacity of pNIPAM-encapsulated gold nanoparticles [18]. Hence, as the incorporation of gold cores modulates the overall gold-hydrogel core-shell size (Table 2), the number of nucleation points seems to have a direct impact on the internal structure homogeneity of the hydrogel shells (and thereby on the shrinkage capacity) by reducing the cross-linking gradient effect for a given polymerization condition, e.g., the concentration of monomer and cross-linker. This behavior is observed in the sample group S1, where a continuous decrease in the shrinking ratio occurs when increasing the number of nucleation points over 8 × 10^12^ NPs (Figure 3).

## 4. Conclusions

This work addressed the critical role of nucleation points on the synthesis of gold-hydrogel core-shell nanoparticles by providing new insights concerning their influence on the modulation of crucial parameters, including the encapsulation yield, size, and shrinkage capacity. The variation in the number of the functionalized gold cores employed in the seeded precipitation polymerization process resulted in the control of the successful encapsulation yield of gold nanoparticles from 15% to 99%. This led a perfectly matched tunability even when different gold core sizes were used at similar particle concentration, demonstrating that the hydrogel encapsulation of gold cores varies in proportion with the number of nucleation points rather than by the overall particle surface. Notably, this new understanding allowed the encapsulation of bimodal gold nanoparticles in the same polymerization process with high control over the encapsulation yield, suggesting that the same principle could be extended to the encapsulation of hydrophobic cores composed of other materials. Likewise, it was observed that the number of nucleation points has a direct effect on both the size and shrinkage capacity of the core-shell nanoparticles. Indeed, raising the number of nucleation points resulted in thinner hydrogel coatings and better shrinkage capacities. These results suggest further investigations are needed to understanding better the effect of nucleation points on the structure of the hydrogel network.

## Figures and Tables

**Figure 1 polymers-13-00373-f001:**
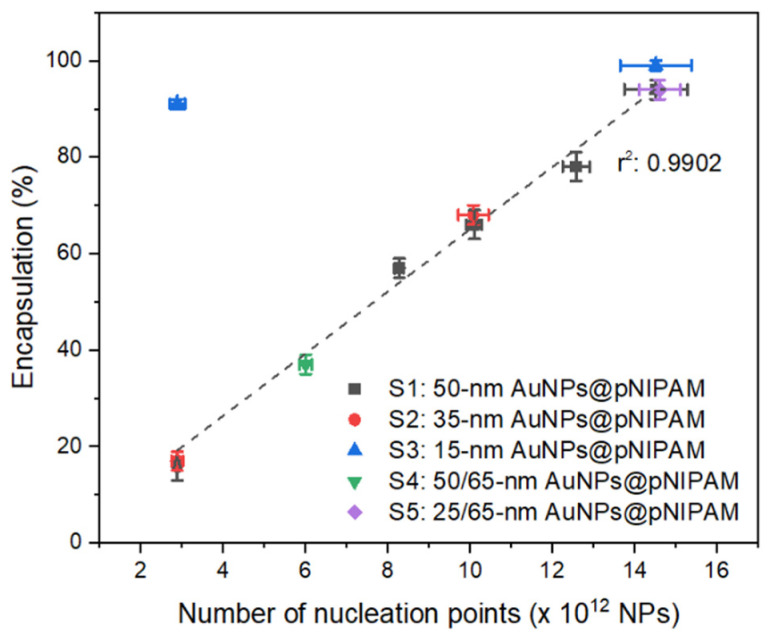
Successful gold cores encapsulation percentages as a function of the number of nucleation points used in the distinct polymerization experiments.

**Figure 2 polymers-13-00373-f002:**
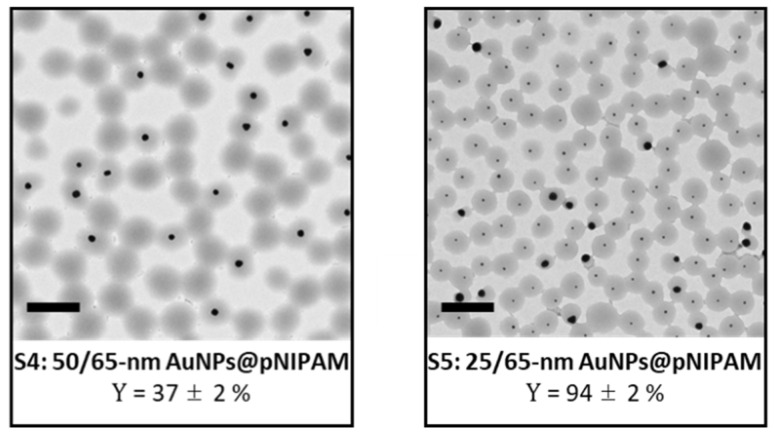
Representative TEM images of both synthesized bimodal samples: (**Left**) S4: 50/65 nm AuNPs@pNIPAM and (**Right**) S5: 25/65 nm AuNPs@pNIPAM. Scale bars are 500 nm.

**Figure 3 polymers-13-00373-f003:**
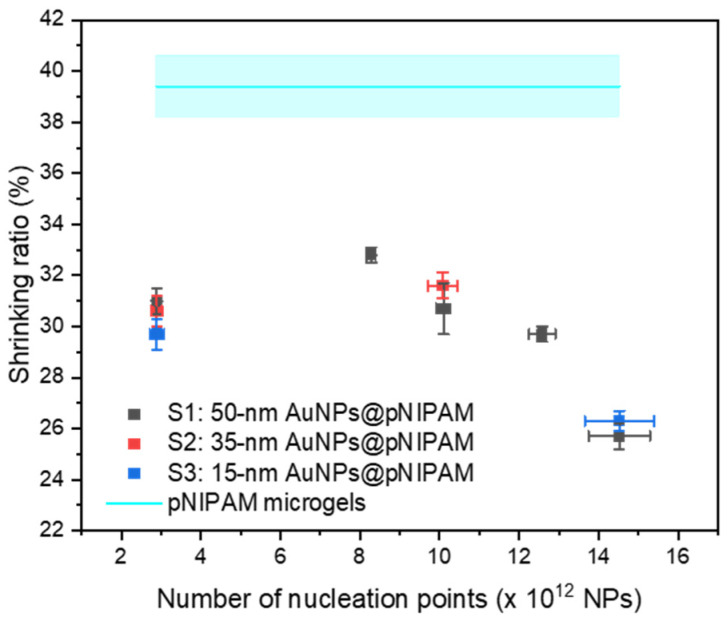
Modulation of the shrinking ratio as a function of the number of nucleation points.

**Table 1 polymers-13-00373-t001:** Summary of the performed syntheses and their resulting successful encapsulation yields ^‡^.

Samples	Number of Nucleation Points (×10^12^)	Overall Particle Surface (cm^2^)	Encapsulation (%)
S1: 50 nm AuNPs@pNIPAM-1	2.9 ± 0.1	305 ± 84	16 ± 3
S1: 50 nm AuNPs@pNIPAM-2	8.3 ± 0.1	599 ± 100	57 ± 2
S1: 50 nm AuNPs@pNIPAM-3	10.1 ± 0.2	1031 ± 290	66 ± 3
S1: 50 nm AuNPs@pNIPAM-4	12.6 ± 0.3	1284 ± 183	78 ± 3
S1: 50 nm AuNPs@pNIPAM-5	14.5 ± 0.8	1353 ± 210	94 ± 2
S2: 35 nm AuNPs@pNIPAM-1	2.9 ± 0.1	138 ± 50	17 ± 2
S2: 35 nm AuNPs@pNIPAM-2	10.1 ± 0.4	366 ± 130	68 ± 2
S3: 15 nm AuNPs@pNIPAM-1	2.9 ± 0.2	29 ± 13	91 ± 1
S3: 15 nm AuNPs@pNIPAM-2	14.5 ± 0.9	241 ± 85	99 ± 1
S4: 50/65 nm AuNPs@pNIPAM	6.0 ± 0.2	583 ± 38	37 ± 2
S5: 25/65 nm AuNPs@pNIPAM	14.6 ± 0.5	1006 ± 100	94 ± 2

^‡^ Each polymerization was performed with its dedicated gold nanoparticle batch to meet the requirements of the total number of nucleation points. Slight variations in core size from batch to batch have been considered for calculating the overall particle surface across samples.

**Table 2 polymers-13-00373-t002:** Summary of the DLS measurements at 15 and 50 °C and the shrinking ratio of each sample.

Samples	Number of Nucleation Points (×10^12^)	D_H_ at 15 °C (nm) ^‡^	D_H_ at 50 °C (nm) ^‡^	Shrinking Ratio (%)
S1: 50 nm AuNPs@pNIPAM-1	2.9 ± 0.1	501 ± 6	339 ± 5	31.0 ± 0.5
S1: 50 nm AuNPs@pNIPAM-2	8.3 ± 0.1	370 ± 3	255 ± 1	32.8 ± 0.3
S1: 50 nm AuNPs@pNIPAM-3	10.1 ± 0.2	391 ± 7	264 ± 8	30.7 ± 1.0
S1: 50 nm AuNPs@pNIPAM-4	12.6 ± 0.3	345 ± 1	230 ± 3	29.7 ± 0.3
S1: 50 nm AuNPs@pNIPAM-5	14.5 ± 0.8	231 ± 4	146 ± 3	25.7 ± 0.5
S2: 35 nm AuNPs@pNIPAM-1	2.9 ± 0.1	378 ± 2	254 ± 6	30.6 ± 0.6
S2: 35 nm AuNPs@pNIPAM-2	10.1 ± 0.4	372 ± 3	253 ± 4	31.6 ± 0.5
S3: 15 nm AuNPs@pNIPAM-1	2.9 ± 0.2	305 ± 5	204 ± 3	29.7 ± 0.6
S3: 15 nm AuNPs@pNIPAM-2	14.5 ± 0.9	188 ± 4	120 ± 1	26.3 ± 0.4
pNIPAM microgels	-	331 ± 6	243 ± 5	39.4 ± 1.2

^‡^ The uncertainties correspond to the standard deviation of the mean hydrodynamic diameter values obtained by DLS (N = 3) and do not represent a measure of the polydispersity of the core-shell particle size distribution.

## Data Availability

The data presented in this study are available on request from the corresponding author.

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
