# Peer review of "Nucleation Points: The Forgotten Parameter in the Synthesis of Hydrogel-Coated Gold Nanoparticles"

_polymers, 2021, doi:10.3390/polym13030373_

Round 1
Reviewer 1 Report
In this work by Sepúlveda et al., the authors made an interesting observation that the concentration of gold cores determines the end effect of the seeded precipitation process. Although this is a communication article, so a certain degree of incompleteness is expected, the presented results appear sound and interesting. Moreover, they definitely fit the scope of Polymers. My recommendation is "minor revision". Upon incorporation of the following suggestions, the article will be ready for publication.
1) It is essential to provide all the parameters needed to reproduce the study. This enables others to verify the findings and build on them. However, some details are missing from the text. Examples:
- "NIPAM, which was purified by recrystallization." (Line 70) - the procedure is not described
- "The gold nanoparticles were functionalized with B-en-A" (Line 92) - the meaning of B-en-A is not specified
- some characterization parameters such as TEM acceleration voltage are also missing
Please carefully screen the experimental section for all the shortcomings (there are more of them) and make the necessary amendments.
2) Distribution histograms in Figs. S1-3 are too small to read.
Author Response
Reviewer 1 :
In this work by Sepúlveda et al., the authors made an interesting observation that the concentration of gold cores determines the end effect of the seeded precipitation process. Although this is a communication article, so a certain degree of incompleteness is expected, the presented results appear sound and interesting. Moreover, they definitely fit the scope of Polymers. My recommendation is "minor revision". Upon incorporation of the following suggestions, the article will be ready for publication.
1) It is essential to provide all the parameters needed to reproduce the study. This enables others to verify the findings and build on them. However, some details are missing from the text. Examples:
- "NIPAM, which was purified by recrystallization." (Line 70) - the procedure is not described.
Revised. In line 70, the following sentence was expanded to complement the information: “N-isopropylacrylamide (NIPAM; Sigma-Aldrich, 97%) was purified by recrystallization in n-hexane prior to use”.
- "The gold nanoparticles were functionalized with B-en-A" (Line 92) - the meaning of B-en-A is not specified
Revised. All abbreviations of compounds are presented in section 2.1. Chemicals (lines 64 - 73). As an example: “Butenylamine hydrochloride (B-en-A; Sigma-Aldrich, 97%)”.
- some characterization parameters such as TEM acceleration voltage are also missing
Revised. The information regarding the accelerating voltage of TEM characterization was added in line 122 “(…) at an accelerating voltage of 100 kV”.
Please carefully screen the experimental section for all the shortcomings (there are more of them) and make the necessary amendments.
Revised. Complementary information was included in section 2.4. Characterization (lines 119 - 132) concerning the preparation of samples for TEM imaging, employed software for image analysis, and the equipment for nanoparticle track analysis measurements.
2) Distribution histograms in Figs. S1-3 are too small to read.
Revised. The size of each size distribution histogram in the electronic supplementary information (ESI) document was adjusted for better visibility.
Reviewer 2 Report
Comments to the author:
In this manuscript, the authors provided new insights for the synthesis of core-shell gold- poly-N-isopropylacrylamide (pNIPAM) nanoparticles regarding the importance of using the concentration of gold cores as a control parameter in the seeded precipitation polymerization process to modulate – regardless of core size – relevant fabrication parameters such as encapsulation yield, particle size and shrinkage capacity. Overall, I would recommend the publication of this contribution because it gives light on the synthetic mechanism of core-shell systems, allowing yields of encapsulation up to 99% for 15 nm gold nanoparticles.
The following are some questions and suggestions for improving their work:
Minor issues:
- Regarding the bimodal systems, could the authors comment of the fact that for the system of 50/65 nm gold nanoparticles a maximum yield of encapsulation of 37% is achieved?
- Could the authors provide SEM characterization of the pNIPAM system with no gold nanoparticles?
- Could the authors comment of why in Figure 3, the control experiment pNIPAM without NPs has different values regarding the nucleation points? It should be just one value, isn’t it? If the authors provide 5 different standard deviations it could be misleading.
Author Response
Reviewer 2:
In this manuscript, the authors provided new insights for the synthesis of core-shell gold- poly-N-isopropylacrylamide (pNIPAM) nanoparticles regarding the importance of using the concentration of gold cores as a control parameter in the seeded precipitation polymerization process to modulate – regardless of core size – relevant fabrication parameters such as encapsulation yield, particle size and shrinkage capacity. Overall, I would recommend the publication of this contribution because it gives light on the synthetic mechanism of core-shell systems, allowing yields of encapsulation up to 99% for 15 nm gold nanoparticles.
The following are some questions and suggestions for improving their work:
Minor issues:
- Regarding the bimodal systems, could the authors comment of the fact that for the system of 50/65 nm gold nanoparticles a maximum yield of encapsulation of 37% is achieved?
The achieved encapsulation yield of 37% for sample S4: 50/65-nm AuNPs@pNIPAM results from the total number of nucleation points (mixture of 50- and 65-nm gold cores) employed in the polymerization process. This value totally agrees with the tendency observed when the number of nucleation points is controlled. In this regard, it is worth noticing that the paragraph starting in line 162 of the manuscript was rewritten at lines 167-168 and 171-172 to clarify the results obtained with bimodal samples S4 and S5.
- Could the authors provide SEM characterization of the pNIPAM system with no gold nanoparticles?
Revised. A transmission electron micrograph of pNIPAM microgels without metallic cores was incorporated in the electronic supplementary information (ESI) document and cited in the manuscript in line 213.
- Could the authors comment of why in Figure 3, the control experiment pNIPAM without NPs has different values regarding the nucleation points? It should be just one value, isn’t it? If the authors provide 5 different standard deviations it could be misleading.
Effectively, the control experiment of pure pNIPAM microgels corresponds to a single synthesis, which leads to only one value of shrinking ratio as presented in Table 2 (line 224). Hence, to avoid any misunderstanding, Figure 3 (line 258) was edited to show the mean shrinking ratio value as a continuous line and its standard deviation as a light color-filled rectangle to allow a direct comparison with the individual shrinking ratios of the hydrogel-encapsulated gold cores.